# Discovery of antitumor lectins from rainforest tree root transcriptomes

**Atip Lawanprasert[1], Caitlin A. Guinan[2], Erica A. Langford[2], Carly E. Hawkins[2], Janna N. Sloand[1], Howard W. Fescemyer[2], Matthew R. Aronson[1], Jacob A. Halle[1], James H. Marden[2,3]\*, Scott H. Medina[1]\***

**1** Department of Biomedical Engineering, Penn State University, University Park, PA, United States of America, **2** Department of Biology, Penn State University, University Park, PA, United States of America, **3** Huck Institutes of the Life Sciences, Penn State University, University Park, PA, United States of America

\* jhm10@psu.edu (JHM); shm126@psu.edu (SHM)

**Data Availability Statement:** All relevant data are within the manuscript and its Supporting Information files, with the exception of microarray data. Data sets generated during microarray

## Abstract

Glycans are multi-branched sugars that are displayed from lipids and proteins. Through their diverse polysaccharide structures they can potentiate a myriad of cellular signaling pathways involved in development, growth, immuno-communication and survival. Not surprisingly, disruption of glycan synthesis is fundamental to various human diseases; including cancer, where aberrant glycosylation drives malignancy. Here, we report the discovery of a novel mannose-binding lectin, ML6, which selectively recognizes and binds to these irregular tumor-specific glycans to elicit potent and rapid cancer cell death. This lectin was engineered from gene models identified in a tropical rainforest tree root transcriptome and is unusual in its six canonical mannose binding domains (QxDxNxVxY), each with a unique amino acid sequence. Remarkably, ML6 displays antitumor activity that is >$10^5$ times more potent than standard chemotherapeutics, while being almost completely inactive towards non-transformed, healthy cells. This activity, in combination with results from glycan binding studies, suggests ML6 differentiates healthy and malignant cells by exploiting divergent glycosylation pathways that yield naïve and incomplete cell surface glycans in tumors. Thus, ML6 and other high-valence lectins may serve as novel biochemical tools to elucidate the glycomic signature of different human tumors and aid in the rational design of carbohydrate-directed therapies. Further, understanding how nature evolves proteins, like ML6, to combat the changing defenses of competing microorganisms may allow for fundamental advances in the way we approach combinatorial therapies to fight therapeutic resistance in cancer.

## Introduction

The outer surfaces of mammalian cells are decorated with a complex array of branched carbohydrates attached to membrane proteins or lipids, collectively referred to as glycans [1]. Structural diversity of these molecules leads to a broad spectrum of biological functions, where glycans play key roles in protein folding, modulation of immune responses and wound healing, to name a few [2, 3]. Dysregulation of glycosylation is also central to the initiation and

studies are available in the BioStudies repository, under accession number S-BSST316: https://www.ebi.ac.uk/biostudies/studies/S-BSST316.

**Funding:** The ML6 sequence was obtained and characterized using support from NSF DEB-1120476 and DEB-1457571, from material covered under export permits and a materials transfer agreement from Panama. This work was generously supported by a Charles E. Kaufman Foundation - New Initiative Award (KA2017-91785). There was no additional external funding received for this study.The funders had no role in study design, data collection and analysis, decision to publish, or preparation of the manuscript.

**Competing interests:** The authors have declared that no competing interests exist.

progression of various human diseases. For example, early stages of cancer are characterized by impairment of normal glycosylation pathways that ultimately produce incomplete glycan structures [4]. Conversely, changes in gene expression in advanced cancers can lead to the *de novo* synthesis of unique glycans [5]. Collectively, this aberrant glycosylation can inhibit immune-mediated destruction of tumor cells and induce signaling pathways that promote carcinogenesis, tumor metastasis and contribute to drug resistance [6]. Despite the link between defective glycosylation and malignancy, the glycome of many tumor types remains largely unknown and, along with it, a wealth of potentially novel targets for therapies [3, 4]. Molecules capable of distinguishing altered expression of glycans on malignant cells represent powerful biochemical tools to probe pathways important to cancer biology, and in some cases also disrupting cell functions to serve as potential therapeutics.

One such class of agents are lectins, which comprise a diverse family of carbohydrate-binding proteins that recognize and bind to glycans with sugar-specificity. Lectins are found in organismal proteomes [7, 8], and secretomes [9, 10] across all kingdoms of life, and have a variation of ligand selectivity that rivals' that of antibodies. In many cases mammalian cell-surface binding of lectins, and subsequent intracellular transport, can modulate a manifold of biologic pathways with remarkable potency [11]. For instance, less than 2 mg of ricin, a ribosome-inactivating lectin produced by the castor bean plant [12], is sufficient to kill an average sized human. A more positive example is the human mannose-binding lectin (MBL), a component of the innate immune system, which is responsible for binding to the high-mannose glycoprotein coat of pathogenic microbes and apoptotic cells to promote their destruction and removal [13].

There is abundant evidence for the therapeutic potential of lectins from diverse sources in nature. For example, a mannose-binding lectin from red-algae has nanomolar inhibitory activity against HIV, hepatitis C, severe acute respiratory syndrome (SARS) coronavirus and *Ebola* viruses, while displaying no deleterious effects on human immune cells and rodent animal models [14]. In one notable study, mice treated with a high-dose of a recombinant mannose-specific lectin survived otherwise fatal *Ebola* viral infection and became immune to virus rechallenge [15]. Interestingly, an individual human's expression level of MBL also affects incidence and severity of certain cancers [16], suggesting that, in addition to their antiviral activity, lectins may also play central roles in tumor prevention and therapy.

With this background in mind, we began to explore the biologic activity of a family of lectins, newly discovered during the course of functional genomic analyses of root specimens from a small sample of tropical rainforest tree seedlings (**Fig 1**) [17]. Using a recombinant lectin expressed from this library we determine how this protein interacts with and potently inhibits human cancer cell lines. These findings may shed light on the glycomic signature of human tumors, identify new vulnerabilities of cancer, and establish a foundation for the future development of novel carbohydrate-targeted lectin therapeutics and diagnostics in oncology.

## Materials and methods

### Materials

Anti6xHIS-FITC was obtained from Abcam (Cambridge, United Kingdom). Dimethyl sulfoxide (DMSO) cell culture grade and Bovine Serum Albumin (BSA) were purchased from Fisher BioReagents (Bellefonte, PA). DMSO spectrophotometric-grade was purchased from Alfa Aesar (Haverhill, MA). Dulbecco Minimum Essential Media (DMEM), Fetal Bovine Serum (FBS), L-Glutamine, Trypsin EDTA and RPMI-1640 were purchased from Corning (Corning, NY). Low Serum Growth Supplement (LSGS), D(+)-Mannose and Medium-106 were purchased from Thermo Fisher Scientific (Bellefonte, PA). Vascular Cell Basal Medium and Endothelial

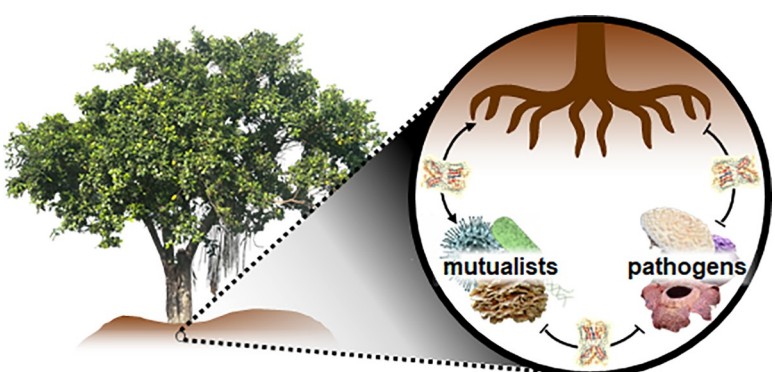

**Fig 1. Roles of microorganismal lectins.** Lectins (highlighted yellow) found in root systems of plants and trees elicit a diverse array of functions. These include defense against pathogens, as well as support for mutualists and symbiotes. Lectins are also key weapons in the arms-race of competing microorganisms.

Cell Growth Kit–VEGF were purchased from ATCC (Manassas, VA). Gentamycin hydrochloride and Ham's F12 media was purchased from VWR (Radnor, PA). 3-(4,5-dimethylthiazol-2-yl)-2,5-diphenyltetrazolium bromide (MTT dye) was purchased from Chem-Impex International (Wood Dale, IL). Paraformaldehyde solution 4% in PBS (PFA) was purchased from Santa Cruz Biotechnology (Dallas, TX). Triton-X100, Tween-20, EmbryoMax Ultrapure Water with 0.1% Gelatin, MEM non-essential amino acid solution, D-(+)-Glucose, Insulin (Recombinant human), Transferrin Apo- (human plasma), hydrocortisone, and sodium bicarbonate were purchased from Millipore-Sigma (Burlington, MA). FITC Annexin V Apoptosis Detection Kit was purchased from BD Pharmingen (San Jose, CA). A549, HeLa, Jurkat, Caco-2, MCF-7, OVCAR-3, NCI/ADR-RES and T24 cell lines were generously provided by the National Cancer Institute (Rockville, MD). Human Dermal Fibroblast (HDF) and Human Umbilical Vein Endothelial cells (HUVEC) were kindly provided by Dr. Yong Wang's laboratory (Penn State, Department of Biomedical Engineering). NL20 cells were kindly provided by Taylor Group in the Division of Thoracic Surgery at the Penn State Hershey Medical Center.

### Lectin sequence identification and selection

In a study described previously [17], we isolated RNA from roots of seedlings from six different species of tropical trees and synthesized cDNA using a poly-A primer. This process produced almost exclusively eukaryotic genes (those with poly-A tails on RNA) from the plant root tissue and associated eukaryotic microbes. We performed *de novo* transcriptome assembly to construct gene models and used a Hidden Markov Matrix procedure to detect deep homology and identify gene models from plants by searching for similar genes in 22 representative high-quality plant genomes. Remaining sequences (N = 268,008 unique sequences) were retained as a heterogeneous collection of genes most likely originating from diverse root-associated eukaryotic microbes, most of which appear to be fungi.

We next searched those genes for deduced protein sequences likely to show anti-biological activity. For this purpose we used a database of bacteriocins (http://bagel2.molgenrug.nl) as Blast query sequences. Searching for bacterial genes in sequences of eukaryotic soil microbes was motivated by the hypothesis that horizontal gene transfer may have occurred. The best match from this search involved the query protein putidacin_L1 (*Pseudomonas sp.*; 31% identical amino acids and 130–144 mismatches out of 230 aligned amino acids, bitscore = 66). This is a weak match, as expected given that we were looking for horizontally transferred genes.

Putadicin_L1 contains two canonical mannose-binding domains (QxDxNxVxY), involved in alpha-D-mannose recognition. This query sequence, and more specifically its mannose-binding domains, aligned to two sequences from *Virola surinamensis* roots, Virola_comp126176_c0_seq1, and Virola_comp114133_c0_seq1. Deduced proteins from these sequences contain 12 and 6 mannose-binding domains, respectively. Nine of the domains in Virola_comp126176_c0 are different from each other, as are all six in Virola_comp114133_c0_seq1. This diversity of binding motifs caught our attention as it suggested that minor changes in a particular glycan on target cells could reduce binding with one particular mannose-binding motif, but not others, making these lections somewhat "evolution proof". Hence, these looked like versatile and possibly useful mannose binding lectins. To our knowledge, high valency of mannose-binding domains and sequence diversity among those domains has not been used to select lectins for testing as reagents, therapeutics, and other applications.

## Lectin expression and purification

Double-stranded DNA was synthesized (IDT gblocks®) using codons optimized for expression in bacteria, with a 5' polyhistidine encoded tag and thrombin-cleavage site, with additional non-coding flanking sequences to facilitate Gibson assembly. The construct was cloned into the pET28b vector, transformed into *E. coli* BL21(DE3) cells, and the insert sequence was verified via Sanger sequencing. A master cell bank was formed and kept at -80˚C for sub-culturing and 5 L IPTG-induced batch fermentation. The harvested cell paste from fermentation was combined with PBS and protease inhibitor cocktail (SigmaFAST™ EDTA free) and disrupted in a microfluidizer. The resulting lysate was then clarified via centrifugation and loaded onto a $Co^{2+}$ column, which bound to the recombinant protein's poly-His tag through ion-exchange chromatography. The protein was eluted with 150 mM imidazole and purified by FPLC fractionation. Purity was verified using standard SDS-PAGE methods. The fractions were dialyzed against PBS to remove remaining imidazole and flowed through an affinity column for endotoxin removal (Sartorius Sartobind®), yielding a verified endotoxin concentration of 0.105 EU/mL (Charles River Laboratories) in a stock containing 2.76 mg/mL of recombinant protein. Aliquots of the lectin were used for experiments within three days of being thawed. The lectin in PBS buffer was colloidally stable up to one week at 4˚C.

## Lectin structural modeling

The primary amino acid sequence for ML6 was searched via BLAST[18] and HHBlits[19] against the SWISS-MODEL template library (SMTL, SMTL version 2019-07-18, PDB release 2019-07-12). This totaled 688 templates found. Each template identified had predicted quality assessed using a QMEAN scoring function from the target-template alignment [20]. The 7 resulting higher quality templates were selected for model building (S1 Table) through Pro-Mod3 and the first model of the selection (3m7j.2.A) was then used to build a three-dimensional representation of the ML6 homology model through the use of PyMOL. The resulting model01.pdb file of ML6 was then exported into PyMOL version 2.3.3 and the sequence was displayed for viewing. Protein measurements were performed using the PyMOL measurement wizard.

## Cell culture

A549, HeLa, Jurkat, MCF-7, OVCAR-3, NCI/ADR-RES and T24 cell lines were cultured in complete growth media comprised of 10% v/v FBS, 2 mM L-Glutamine, and 0.05 mg/mL gentamycin in RPMI-1640. Caco-2 cell line was cultured in 10% v/v FBS, 2 mM L-Glutamine, and 0.05 mg/mL gentamycin in DMEM. HDF cell line was cultured in 2% LSGS in Medium-106.

HUVEC cells were cultured in Vascular Cell Basal Medium supplemented with 5 ng/mL rhVEGF, 5 ng/mL rh EGF, 5 ng/mL rh FGF basic, 15 ng/mL rh IGF-1, 10 mM L-glutamine, 0.75 Units/mL Heparin sulfate, 1 μg/mL Hydrocortisone hemisuccinate, 2% FBS, and 50 μg/mL ascorbic acid. The NL20 (CRL-2503) cell line was cultured in Ham's F12 media, supplemented with 1.5 g/L sodium bicarbonate, 2.7 g/L glucose, 2.0 mM L-Glutamine, 0.1 mM non-essential amino acids, 0.005 mg/mL insulin, 10 ng/mL epidermal growth factor, 0.001 mg/mL transferrin, 500 ng/mL hydrocortisone, and 4% FBS. Cells were incubated at 37˚C with 5% $CO_2$.

## Circular dichroism

Lectin samples were diluted in CD buffer (7.5 mM sodium phosphate dibasic, 2.5 mM sodium phosphate monobasic, pH 7.4) to a final protein concentration of 20 μM. The sample solutions were then added to the 10 mm path length quartz cell before collecting CD spectra using a J-1500 Circular Dichroism Spectrometer (JASCO; Oklahoma City, OK). The ellipticity (mdeg) measurements were done at 25˚C with data collected over a wavelength range of 180 nm to 260 nm. Three experimental replicates were performed, with three intra-experimental replicates, for each sample.

## Anticancer screening

In 100 μL of the appropriate culture growth medium, A549, HeLa, Jurkat, and T24 cell lines were seeded in 96-well plates at 2000 cells/well. MCF-7, OVCAR-3, Caco-2, HDF, HUVEC and NL20 cell lines were seeded in 96-well plates at 5000 cells/well. For HUVEC cells the 96-well plate was treated with 0.1% Gelatin prior to seeding. The cells were allowed to adhere under normal culture conditions for 24 hours before treatment. For plates seeded with Jurkat, gentle centrifugation (3000 rpm, 5 minutes) was used in each step to remove the supernatant. The growth medium was then replaced by 100 μL of the appropriate culture growth medium containing $10^{-10}$–1 mg/mL of ML6, prepared via serial dilution. Treatment solutions of Doxorubicin and Paclitaxel were prepared by serial dilution in growth medium to achieve concentrations of $10^2$–$10^{-3}$ μM and $10^0$–$10^{-5}$ μM, respectively. Blank media and 20% DMSO were included as negative and positive controls, respectively. After 48 hours of incubation the treatment supernatant was removed, and 100 μL of a 0.5 mg/mL MTT solution in media added to each well. The plate was then incubated at 37˚C for 2 hours before removing the unreacted MTT solution. Spectrophotometric-grade DMSO (100 μL) was added to each well to solubilize the formazan product. After 15 minutes incubation in a 37˚C oven to dissolve, the absorbance was read at 540 nm using a microplate reader (Cytation 3, BioTek; Winooski, VT). Percent viability was calculated using the following equation: (Absorbancetreatment−Absorbance_positive)/(Absorbancenegative—Absorbance_positive) x 100%. Three experimental replicates were performed, with three intra-experimental replicates, for each sample. $IC_{50}$ values were calculated by fitting cytotoxicity curves using nonlinear regression in GraphPad Prism software. In parallel, the cellular morphology was assessed using an inverted microscope (Olympus; Tokyo, Japan). The micrographs were taken at 20X magnification with a phase contrast filter.

For competitive inhibition studies, A549 cells were seeded at 2000 cells/well in a 96-well plate and allowed to adhere overnight. The growth medium was then removed and replaced by 100 μL of an ML6 treatment solution prepared by serially diluting the protein to $10^{-2}$–$10^{-7}$ mg/mL in media. Free D-mannose was added to each solution at 0 – 1mM concentrations before addition to cells. Blank media and 20% DMSO containing the same D-mannose concentrations were included as negative and positive controls, respectively. After 48 hours, the treatment supernatant was removed and cell viability assayed as described above.

## Cell apoptosis assay

A549 cells were seeded in 24-well plates at 200,000 cells/well and allowed to adhere under normal culture conditions for 24 hours before treatment. Supernatant in treatment wells was removed and replaced by media containing 1pM of ML6 before incubating for 1 hour or 14 hours. Cells treated with blank media, or 20% DMSO for 1 hour, served as negative and positive controls, respectively. Cells were washed with cold PBS twice before removal via addition of 200μL Trypsin-EDTA. After 5 minutes of incubation, 300 μL of PBS was added and the cells solution was retrieved, centrifuged (3000 rpm, 5 minutes) to remove supernatant, and pellet resuspended in 100 μL 1X Binding Buffer. To stain treated cells, 5 μL of FITC Annexin V and 5 μL of Propidium Iodide (PI) solutions were added to each sample tube, before gentle vortexing to mix and allowed to incubate in the dark at room temperature for 15 minutes. After incubation, 400 μL of 1X Binding Buffer was added to each sample before analysis of FITC and PI fluorescence using a BD LSRFortessa Cytometer (BD Bioscience; San Jose, CA). To adjust for FITC and PI fluorescence overlap compensation controls were retrieved and using the same protocol as above. During data collection events were gated with respect to the unstained control and quadrant scatter plots produced for analysis.

## Cell aggregation assay

Suspended A549 cell lines were diluted in RPMI-1640 to prepare a cell stock with final concentration of $10^5$ cells/mL. ML6 (10μM final concentration) and cell solutions were added in equal volumes to a 12-well plate to a final volume of 1 mL. For the negative control, 1X PBS was added to the cell stock at the same volume of ML6. Cells were incubated for 30 minutes on a heated shaker (180 rpm, 37˚C), allowed to rest for an additional 15 minutes and then subjected to phase contrast microscopy. Average cluster area was determined by measuring sizes of cell aggregates (>3 cells) using ImageJ software.

## Confocal microscopy

A549, MCF-7 or NL20 cell lines were seeded in a chamber slide (LabTek™ II Chamber Slide™ 4-well, Thermo Fisher Scientific; Waltham, MA) at 20,000 cells/well in 500 μL complete growth media and allowed to adhere overnight. To prepare the treatment solution, ML6 was diluted in 500 μL complete growth media to a final protein concentration of 10 μM. The growth media was replaced by the treatment solution and incubated for 1 hour or 24 hours. Blank media was included as a negative control. Following treatment, cells were washed 3 times with PBS then fixed and permeabilized with 4% PFA in PBS and 0.1% Triton-X100 in PBS for 15 minutes at room temperature, respectively. After that, cells were washed three times with PBS to remove the remaining fixative and permeabilizing reagents. Cells were then blocked with 1% BSA and 0.1% Tween-20 in PBS for 24 hours at 4˚C. The blocking solution was removed and the antibody tagging solution (1:1000 FITC-tagged Anti-6XHIS in PBS containing 1% BSA) was added and incubated in the dark at room temperature for 1 hour. Cells were then washed with PBS to remove unbound antibody. Following the preparation, the chamber walls were removed, and the slides mounted using ProLong Diamond Antifade Mountant with DAPI (Thermo Scientific; Waltham, MA). The sample slides were left to cure at 4˚C overnight before imaging. Imaging was performed using a FLUOVIEW 1000 Confocal Microscope (Olympus; Tokyo, Japan) outfitted with 359 nm and 489 nm lasers for DAPI and FITC fluorescent, respectively. The micrographs were post-processed and relative fluorescence semi-quantitatively analyzed via ImageJ. Average fluorescence per cell was normalized by untreated cell control, shown as mean ± SEM for n = 20 cells per condition.

## Glycan microarray

Microarray screening studies were performed using a RayBio® Glycan Array 300 Kit (Ray-Biotech, Peachtree Corner, GA), which consists of two blocks of 300 synthetic glycan spots, along with relevant control spots, each 3-fold replicated. The manufacturers protocol was followed to perform the sandwich-based method for antibody-based detection of ML6 captured on the array spots. Lectin was diluted to 0.1 mg/ml in manufacturer-provided diluent, and then incubated overnight at 4˚C to allow hybridization to the experimental array block. Detection of ML6 hybridized spots was enabled using 6x-His Tag Monoclonal Antibody (mouse IgG1, 4E3D10H2/E3) labeled with Alexa Flour 555 (ThermoFisher Scientific, #MA1-135-A555). The antibody was used at 1:500 and applied to both the experimental (lectin treated) and reference blocks (only Ab treated). Detection (imaging) of fluorescing spots was performed with a GenePix 4000B scanner with PMT of 750 and resolution of 5. Median fluorescent signal intensity, after local background subtraction, was normalized according to the manufacturers recommendation and used in t-tests to determine binding significance of ML6 protein (see S2 Table). Data sets generated during the current study are available in the BioStudies repository, under accession number S-BSST316: https://www.ebi.ac.uk/biostudies/studies/S-BSST316.

# Results

## ML6 discovery, ecology and structural characterization

Within libraries of poly-adenylated gene models (i.e. eukaryotic) obtained from *de novo* transcriptomes of six species of tropical tree seedling root samples [17], we performed bioinformatic searches to discover potentially bioactive proteins. Among these we identified a diverse library of >80 novel lectins. Notably, species-rich rainforest root communities [17][21] comprise mutualists, saprophytes, pathogens [22], and even predators (fungi that use lectins to trap and eat nematodes [23]). This diversity suggests that some of the identified proteins in this community may be involved in between-species interactions [24], and potentially serve as components of defensive or offensive systems that may have co-evolved with rival organisms. The first lectin we examined for functional properties was based on a sequence obtained from the roots of *Virola surinamensis* (*Myristicaceae*). A double-stranded DNA was synthesized (IDT gblocks®) using codons optimized for bacterial expression and produced in *E. coli* as a recombinant protein (see methods).

This 28,058 Da protein (**Fig 2A,** S1 Fig), named ML6 (Mannose-binding Lectin 6), contains six unique instances of the canonical QxDxNxVxY domain (where X can be any amino acid) characteristic of bulb-type lectin mannose-binding domains (**Fig 2A**). Isothermal titration calorimetry experiments confirm that recombinant ML6 binds to monosaccharide mannose haptens with high affinity ($K_d = 1.5$ μM; S2 Fig). Although the species origin of this lectin is unknown, a mycologic assay found that it inhibits the growth of the pathogenic plant fungus *Mycosphaerella zeae* (S3 Fig). This antifungal activity suggests ML6 may be produced by a mutualist that defends tree seedlings from pathogenic fungi or competes in some other ecological context. Interestingly, ML6 has only 39% overall amino acid identity to known proteins (NCBI protein and translated nucleotide databases), while the additional 80 lectins in our library have <60% amino acid identity with sequences in any database. This low identity suggests ML6, along with the other lectins in the library, come from species fairly distant from taxa with characterized genomes, and thus may have unique protein conformations and potentially novel biochemical functions.

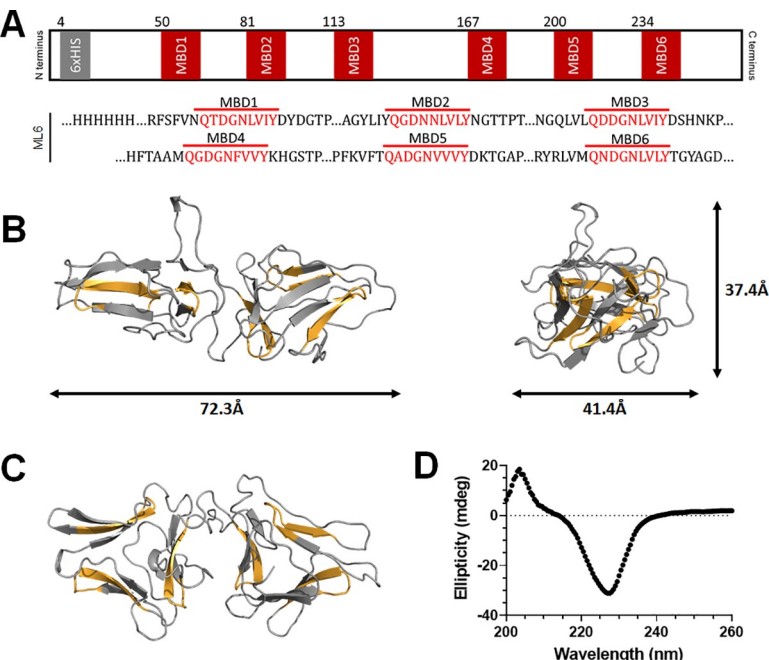

**Fig 2. Sequence and structure of recombinant ML6.** (**A**) Schematic representation of the ML6 sequence (*top*) with unique instances of mannose-binding domains identified (*bottom*, red). (**B**) Predicted computational structure of ML6 using the SWISS model (bacteriocin LlpA reference) viewed from the front (*left*) and side (*right*). (**C**) Top-down view of ML6 emphasizing the central "C" cup. Mannose-binding domains are highlighted yellow. (**D**) Circular dichroism spectrum of recombinant ML6 protein.

To better understand the structure of ML6, we performed predictive protein homology modeling using Bacteriocin LlpA as a reference (S1 Table and S4 Fig). Bacteriocin LlpA is an antibacterial protein that adopts a fold very similar to that of mannose-binding lectins [25]. Models show that ML6 is predicted to possess a β-sheet rich globular structure (**Fig 2B**) with a central "C" cup fold (**Fig 2C**) typical of many lectins [8, 26]. This cleft is flanked by two symmetrical arms each containing two mannose-binding domains, as well as an additional pair of binding surfaces in the central aperture (highlighted yellow in Fig 2B and 2C). Circular dichroism spectroscopy of recombinant ML6 confirms a β-sheet rich secondary structure defined by a single minima in ellipticity at 228 nm (**Fig 2D**).

These physiochemical features of ML6 are roughly similar to that of many legume lectins, which generally range in molecular weight from 25–30 kDa and are characterized by dome-shaped globular structures composed largely of antiparallel β-sheets [8]. However, ML6 exhibits a number of unique and important differences. Most known β-protein lectins display 1–3 monosaccharide binding sites, and associate into homo-dimers, homo-tetramers, or homo-octamers to increase their effective valency [26–28]. These arrangements enhance the capacity, potency, and specificity of lectins to bind their sugar targets, particularly complex *N*-linked glycans [29]. Uniquely, ML6 possesses six mannose-binding domains, all of which have divergent sequences relative to each other and are spaced in close proximity (22–25 amino acids separate domains; **Fig 2A**). This domain number and arrangement are in stark contrast to well-known plant mannose-binding lectins like *Galanthus nivalis* agglutinin 1 and 2 (GNA-1/ 2), which possess two mannose-binding domains in the monomer that are separated by ~50 residues [30, 31]. Thus, ML6 is unusual in its ecological and taxonomic (albeit unknown) origin, monomeric composition, and diversity of mannose-binding domain sequences.

## Anticancer activity

Numerous mannose-binding lectins, particularly those of the GNA family, have been reported to induce apoptosis in cancer cells [32]. For example, garlic bulb (*Allium sativum*) lectins were found to have inhibitory effects on lymphoma cells at concentrations >0.5 mg/mL [33]. Similarly, lectins isolated from the plant *Clematis montana* displayed potent antineoplastic activity towards human breast, ovarian, and liver cancer cell lines [34]. With this background in mind, we screened the toxicity of ML6 against a panel of human cancer cells and normal cell line controls. Results (**Fig 3A** and **Table 1**) demonstrate that ML6 elicits picomolar toxicity toward five of the tested cell lines, including A549, OVCAR-3, HeLa, T24 and Caco-2. Remarkably, for the most sensitive cell line in the panel, A549, ML6 was active even at femtomolar concentrations ($IC_{50}$ = 40 fM; $1.2 \times 10^{-9}$ mg/mL). Competitive inhibition assays demonstrate that the activity of ML6 is critically impaired upon co-incubation with excess mannose (S5 Fig), suggesting mannose-binding is fundamental to ML6's mechanism of action. Active concentrations of ML6 yielded a stark change in A549 cell phenotype after 24 hours of incubation, characterized by the presence of large intracellular vacuoles and contraction of the cell membrane (**Fig 3B**, Top). These morphologic changes were not observed for the ML6-insensitive MCF-7 cell line treated under similar conditions (**Fig 3B**, Bottom). Taken together, these results support our assertion that ML6 avidly interacts with monosaccharide mannose haptens, and suggests that binding to mannose residues on cell surface glycans is a prerequisite for ML6-mediated anticancer activity.

Interestingly, all of the ML6-sensitive cell lines represent human epithelial cancer cells derived from mucosal tissues (e.g. lung, ovarian, cervical, bladder and colon). Conversely, ML6 was found to be inactive towards both of the tested non-mucosal cancer cell lines (MCF-7, Jurkat), as well as three non-cancerous controls (**Fig 3A** and Table 1). Calculating selectivity

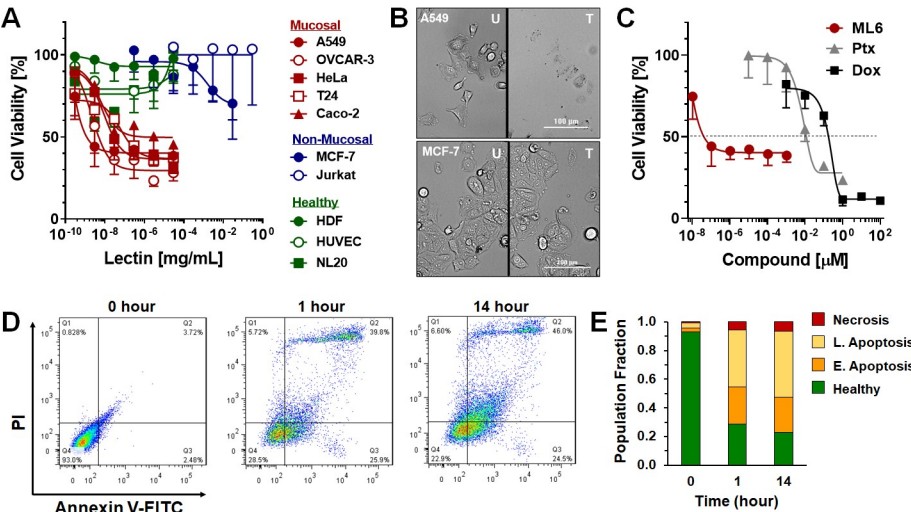

**Fig 3. ML6 anticancer activity and mechanism of action.** (**A**) Cytotoxicity profile of ML6 towards a panel of mucosal (red) and non-mucosal (blue) cancer cell lines, as well as non-cancerous healthy cells (green) as controls. Curves were fit using a non-linear regression analysis to calculate $IC_{50}$ values (see Table 1). (**B**) Brightfield micrographs of ML6-sensitive A549 (top) or the ML6-insensitive MCF-7 (bottom) cell line before (U) and after (T) treatment for 24 hours in the presence of 1nM lectin. Scale bar = 100 μm. (**C**) Representative toxicity curves comparing the anticancer activity of ML6 (red) to the standard chemotherapeutics Ptx (grey) and Dox (black) in A549 cells. Dashed line represents the $IC_{50}$. (**D**) Representative scatter plots from flow cytometric PI/Annexin V-FITC apoptosis assays. ML6-sensitive A549 cells line were treated with 1 pM of ML6 for 1 or 14 hours before analysis. (**E**) Quadrant quantification of flow cytometry data defining the healthy cell population from those in early and late apoptosis, or necrosis, as a function of lectin incubation time.

indices for each cell line (SI; **Table 1**) reveals that ML6 is >100–10,000 times more active towards mucosal cancers than non-malignant cells. This mucosa-specific activity of the lectin is perhaps not surprising given the dense glycocalyx of mucosal epithelium, where surface displayed glycans operate as key intermediaries between cells of the respiratory, urogenital and gastrointestinal tracts and the external environment. For example, mucin-type glycoproteins expressed on epithelia provide important defenses against toxins and pathogenic microorganisms [35]. Abnormal expression of these mucins is a common phenotype of epithelial malignancies, where they play central roles in cancer pathogenesis [36, 37].

Potency of ML6 was next compared to the standard chemotherapeutics Paclitaxel (Ptx) and Doxorubicin (Dox) in the lectin sensitive A549 lung carcinoma cell line. Results in **Fig 3C** show that ML6 ($IC_{50}$ = 4 x $10^{-8}$ μM) is >$10^6$ times more active than Dox ($IC_{50}$ = 0.2 μM) and >$10^5$ more toxic than Ptx ($IC_{50}$ = 0.01 μM). This potent and selective anticancer activity suggests that ML6 is capable of distinguishing altered expression of glycans on malignant mucosal cells and exploits these unique vulnerabilities to disrupt cell function. To evaluate the mechanism and kinetics of action for ML6, we next performed time-dependent flow cytometry assays using the A549 cell line. Results in **Fig 3D & 3E** show that at 1 pM concentrations, ML6 is able to rapidly potentiate early apoptotic signaling pathways, and after 14 hours cells predominately adopt a late apoptotic phenotype.

Curiously, we observed significant batch-to-batch variation of ML6 activity between recombinant expressions (S6 Fig), which was not correlated to significant changes in lectin secondary structure (S7 Fig). In ongoing studies we are testing the sensitivity of ML6 to physical perturbations that may be induced during isolation and purification of the protein that potentially lead to the observed batch-to-batch variance, similar to other recombinantly expressed lectins [38].

## Identifying the glycan targets of ML6

To test our assertion that ML6 recognizes and binds to aberrant cell-surface glycans we performed fluorescent confocal microscopy studies to monitor the time-dependent lectin interactions with cancer cells. Confocal micrographs (**Fig 4**) show that ML6 (the potently anti-cancer batch 1) rapidly localized to the membrane of the lectin-sensitive A549 cell line before being internalized (Fig 4, 1 hour; note punctate and diffuse intracellular fluorescence of the lectin). This decoration of the cell surface by lectins did not result in significant cell-cell agglutination

**Table 1. Anticancer activity of ML6 against a cancer cell line panel.**

| Cell Line | Origin | $IC_{50}$ [mg/mL] | $IC_{50}$ [pM] | SI[a] |
|---|---|---|---|---|
| A549 | Lung Cancer | 1.2 x$10^{-9}$ | 0.04 | 9x$10^4$ |
| OVCAR-3 | Ovarian Cancer | 4.9 x$10^{-9}$ | 0.18 | 2x$10^4$ |
| HeLa | Cervical Cancer | 2.6 x$10^{-8}$ | 0.93 | 4x$10^3$ |
| T24 | Bladder Cancer | 1.1 x$10^{-7}$ | 3.93 | 908 |
| Caco-2 | Colorectal Cancer | 6.7 x$10^{-7}$ | 23.93 | 149 |
| MCF-7 | Breast Cancer | NA | NA | - |
| Jurkat | Lymphocytes | NA | NA | - |
| HDF | Human Dermal Fibroblasts | NA | NA | - |
| HUVEC | Human Umbilical Vein Endothelium | NA | NA | - |
| NL20 | Normal Lung Epithelium | NA | NA | - |

[a]Selectivity index, SI = Max. tested conc. in healthy cells/$IC_{50}$ cell line.

NA = Not active ($IC_{50}$ could not be calculated) at the tested concentration range

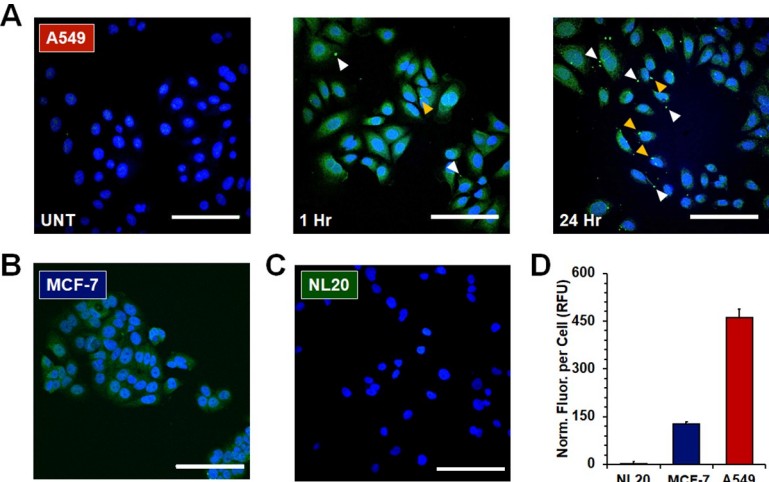

**Fig 4. Subcellular localization and specificity of the ML6 lectin.** (**A**) Merged fluorescent micrographs of A549 cells before (UNT) and after treatment with ML6 for 1 hour and 24 hours. (blue = nucleus, green = lectin). White and yellow arrows highlight punctate regions of ML6 localized to either the cell or nuclear membrane, respectively. Representative micrographs of ML6 localization to the lectin-insensitive (**B**) MCF7 and (**C**) NL20 cell lines after 24 hours of incubation shown as controls. Scale bars = 100 μm. Antibody control images can be found in S9 Fig. (**D**) Normalized cellular fluorescence after 24 hours of incubation with ML6 (shown in relative fluorescence units; RFU).

(S8 Fig). After 24 hours noticeable accumulation of ML6 at the cell surface (white arrows) and nuclear membrane (yellow arrows) was observed. These results suggest a stepwise progression of ML6 penetration into lectin-sensitive tumor cells (cell surface → cytoplasm → nuclear membrane) and interaction with the nuclear envelope, perhaps extending inside the nucleus.

To test the cell-specificity of ML6 binding we performed similar microscopy experiments using the insensitive cancerous and non-cancerous cell lines MCF-7 and NL20, respectively (**Fig 4B & 4C**). These studies demonstrate that ML6 preferentially localizes to the lectin-sensitive cell line (A549) compared to in-sensitive controls (MCF-7, NL20), with a >3-fold increased avidity (**Fig 4D**). This cell-dependent surface binding may offer one explanation as to why ML6 elicits its cytotoxic action against certain cell lines tested in our panel (**Fig 3A**).

We next employed a glycan microarray to characterize the lectin's saccharide specificity. This was performed by hybridizing ML6 to an array of 300 different glycans that encompass both synthetic polysaccharides as well as natural *N*-glycans, glycolipids, human milk oligosaccharides, and tandem epitopes. This analysis revealed high-specificity of the lectin towards five short, mannose-rich *N*-glycans (**Fig 5A** and S2 Table). Among these putative targets is a conserved structural subunit: GlcNAc-β-1,2-Man-α-1,3-Man-β-1,4-GlcNAc-β-1,4-GlcNAc. Given that this carbohydrate pentamer is present in the stem of all human glycans, our results suggest ML6 does not target a unique tumor glycan, but more likely that the accessibility of this motif is increased on aberrant and incomplete glycans overexpressed on the surfaces of malignant cells. Further supporting this assertion is the fact that ML6 did not bind to glycans with identical carbohydrate backbones, but which were terminated with sialic acid (**Fig 5B**). Sialic acids are typically found terminating the branches of fully formed glycans and have been reported to 'hide' mannose antigens on the surfaces of cells from immune-activating MBLs [39]. Taken together this suggests that ML6 may differentiate healthy and malignant cells by binding GlcNAc-β-1,2-Man-α-1,3-Man-β-1,4-GlcNAc-β-1,4-GlcNAc units on naïve and incomplete glycans produced by mis-regulated glycosylation pathways in tumors. This conserved motif of *N*-glycans may also be more sterically shielded on 'healthy' hyperbranched glycan structures capped with terminating sialic acid residues. In addition, ML6 did not bind to a

short glycan in which mannose disaccharides are joined through an α6-linkage (see fourth entry from the top in **Fig 5B**), while a similar backbone with α3-linked subunits was a target for the lectin (see fourth entry from the top in **Fig 5A**). This glycan specificity demonstrates that stereochemical orientation of the divalent mannose moiety in the carbohydrate backbone is also integral to ML6 binding activity.

## Discussion

Although many types of environmental samples and diverse organisms have been examined for bioactive lectins [40–43], microorganisms and terrestrial tropical habitats are under-sampled [44, 45]. Our results indicate that there is a heretofore untapped reservoir of potentially useful lectins to be discovered in this fashion. Our approach is a notable departure from previous efforts as we used *de novo* transcriptome assembly, bioinformatic screening, and DNA synthesis to bypass constraints of protein purification or focal gene approaches [46]. Using such methods in combination with bioinformatic analyses, oligonucleotide synthesis, and recombinant protein production can greatly increase the ease of discovery and testing of novel lectins and other useful proteins.

The first lectin synthesized using this approach, ML6, displayed both anti-fungal and anti-cancer activity, with follow-up characterization of the latter. Confocal microscopy experiments demonstrate that ML6 occupies several extra- and intra-cellular domains in lectin-sensitive cell lines, including at the membrane, dissemination in the cytoplasm, and localization to the nuclear envelope. This distribution pattern may indicate both surface and intracellular locations where ML6 elicits pleiotropic anticancer effects. For example, programmed cell death occurs through extrinsic or intrinsic pathways: death-receptor (extrinsic) mediated apoptosis triggered by the ligation of Fas or other plasma membrane receptors, and a mitochondria-dependent (intrinsic) pathway induced by the release of cytochrome c. A number of recent studies have shown that plant lectins can modulate extrinsic apoptosis signaling in cancer cells by activating cell surface death receptors that include TNF, TRAIL and FasL [47]. Other plant lectins, such as the legume derived ConA, potentiate release of cytochrome c from mitochondrial membranes to trigger intrinsic apoptosis in melanoma and liver cancer cells [48, 49]. Given these examples, it is possible that ML6 acts similarly to ligate cell-surface death receptors

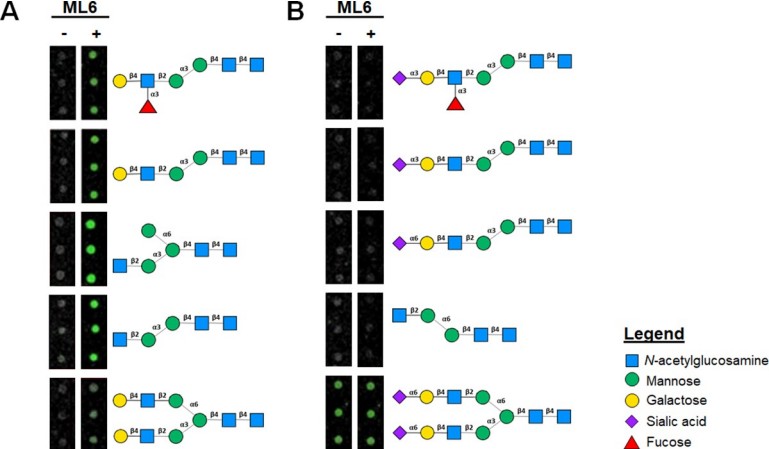

**Fig 5. *N*-glycan binding specificity of ML6.** (**A**) *N*-linked polysaccharides on the glycan microarray to which ML6 avidly bound. (**B**) Glycans derived from a similar carbohydrate backbone, but which did not show ML6-specific recognition. Fluorescent images show triplicate microarray spots of each glycan that were treated in the absence (-) or presence (+) of ML6.

or interact with intracellular mediators to stimulate extrinsic and/or intrinsic apoptotic signaling. This assertion is collectively supported by our data showing that ML6 consists of a β-sheet rich structure (**Fig 2**), like that of many antitumor legume lectins, elicits its cytotoxic effects at incredibly low concentrations (**Table 1**) and induces apoptosis in cells over a time period commensurate with its membrane and cytoplasmic localization (**Figs 3 & 4**).

In addition to its potential to directly induce extrinsic and/or intrinsic apoptosis, ML6 may also bind glycan intermediates in the canonical *N*-glycosylation pathway to alter glycoprotein folding and potentiate cell stress pathways. In fact, microarray data (**Fig 5**) demonstrates that ML6 specifically binds to mannose-rich polysaccharide intermediates central to the eukaryote *N*-glycan biosynthesis pathway [50]. Intracellular abundance of this glycan in the pathway is altered by the Man2a1 enzyme, as well as substrates, co-factors, and nearby enzymes in the biosynthetic process [51]. Recent studies have identified a link between mutations affecting Man2a1 and tumorigenesis [52, 53], adding to a growing body of evidence indicating abnormal *N*-glycosylation processes can drive malignancy. Given this link, there is clinical interest in developing protein-based therapies in oncology that target and inhibit *N*-glycosylation pathways [54]. Yet, the development of therapeutic antibodies against *N*-linked glycans remains challenging due to their limited valency, insufficient sensitivity and poor intracellular penetration [54]. Novel and highly specific mannose-binding lectins, like ML6, which can access intracellular compartments may therefore represent pioneering new therapeutic tools for diseases where *N*-glycosylation has gone awry.

## Supporting information

**S1 Fig. SDS-PAGE gel of the purified recombinant ML6 lectin.**
(DOCX)

**S2 Fig. ITC binding.** Representative isothermal titration calorimetry data of ML6 (163 μM) in the presence of a 10-fold excess of mannose haptens (1.63 mM). $K_d$ = 1.5 μM.
(DOCX)

**S3 Fig. Anti-mycologic activity of ML6.** Hyphae of a plant pathogenic fungus, *Mycosphaerella zeae*, grow profusely in (**A**) untreated media, but are (**B**) stunted following 24 hour treatment with ML6 (500nM). Hyphae remained stunted for 19 days in the presence of ML6, after which the experiment was ended.
(DOCX)

**S4 Fig. Report of target-template model chosen for ML6 homology model build.**
(DOCX)

**S5 Fig. Competitive inhibition of ML6 activity.** Viability of A549 cells in the absence (dark red) or presence of free mannose haptens at a low (0.1 mM, red) and high (1 mM, pink) concentration of the sugar in culture media.
(DOCX)

**S6 Fig. Batch-to-batch variance of ML6 potency.** $IC_{50}$ for three batches of recombinantly-expressed ML6 protein against the A549 lung carcinoma cell line. [†]$IC_{50}$ could not be reached over the tested concentration range.
(DOCX)

**S7 Fig. Circular dichroism analysis of ML6.** CD spectra for batch 1 (•), 2 (■), and 3 (▲) of the recombinantly-expressed ML6 protein.
(DOCX)

**S8 Fig. ML6-mediated cell agglutination.** Average cluster area of cell aggregates measured from microscopy images of A549 cells before (black bar) and after (open bar) a 1 hour incubation with 10 μM ML6.
(DOCX)

**S9 Fig. Antibody staining control images.** Fluorescent micrographs of A549 cell controls stained with antibody (1:1000 FITC-tagged Anti-6XHIS) in the absence of ML6 pre-treatment.
(DOCX)

**S1 Table. Templates selected for ML6 model building.**
(DOCX)

**S2 Table. Glycan array fluorescent signal intensity.**
(DOCX)

## Acknowledgments

We acknowledge and thank the Huck Institutes Microscopy, X-Ray Crystallography, and CSL-Behring Fermentation Facilities at Penn State, University Park, PA for assistance with confocal microscopy, CD spectrophotometry, and protein production and purification. We thank Dr. Heather Feaga for assistance in preparation of the ML6 G-block expression construct. The ML6 sequence was obtained with export permits and a materials transfer agreement from Panama.

## Author Contributions

**Conceptualization:** James H. Marden, Scott H. Medina.

**Data curation:** Atip Lawanprasert, Caitlin A. Guinan, Erica A. Langford.

**Formal analysis:** Atip Lawanprasert, Caitlin A. Guinan, Erica A. Langford, Howard W. Fescemyer, James H. Marden.

**Funding acquisition:** James H. Marden, Scott H. Medina.

**Investigation:** Atip Lawanprasert, Caitlin A. Guinan, Erica A. Langford, Carly E. Hawkins, Janna N. Sloand, Howard W. Fescemyer, Matthew R. Aronson, Jacob A. Halle, James H. Marden, Scott H. Medina.

**Methodology:** Atip Lawanprasert, Caitlin A. Guinan, Erica A. Langford, Carly E. Hawkins, Janna N. Sloand, Howard W. Fescemyer, James H. Marden, Scott H. Medina.

**Project administration:** James H. Marden, Scott H. Medina.

**Resources:** James H. Marden, Scott H. Medina.

**Supervision:** Scott H. Medina.

**Writing – original draft:** Atip Lawanprasert, James H. Marden, Scott H. Medina.

**Writing – review & editing:** Caitlin A. Guinan, Erica A. Langford, Janna N. Sloand, Howard W. Fescemyer.

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
