## [Decision Letter · Decision Letter 0]

21 Nov 2019

PONE-D-19-29568

Discovery of antitumor lectins from competing microorganismal ecologic niches

PLOS ONE

Dear Prof. Medina,

Thank you for submitting your manuscript to PLOS ONE. After careful consideration, we feel that it has merit but does not fully meet PLOS ONE’s publication criteria as it currently stands. Therefore, we invite you to submit a revised version of the manuscript that addresses the points raised during the review process. Please note that both reviewers suggest that the ms is still preliminary and novel experiments are critically required for the revised version of the ms.

We would appreciate receiving your revised manuscript by Jan 05 2020 11:59PM. To enhance the reproducibility of your results, we recommend that if applicable you deposit your laboratory protocols in protocols.io, where a protocol can be assigned its own identifier (DOI) such that it can be cited independently in the future. For instructions see: http://journals.plos.org/plosone/s/submission-guidelines#loc-laboratory-protocols

We look forward to receiving your revised manuscript.

Kind regards,

Roger Chammas, M.D, Ph.D

Academic Editor

PLOS ONE

Journal Requirements:

"This work was supported by a Charles E. Kaufman Foundation - New Initiative Award (KA2017-91785; https://www.kauffman.org/), awarded to S.H.M and J.H.M. The funders had no role in study design, data collection and analysis, decision to publish, or preparation of the manuscript.".

i) Please provide an amended statement that declares *all* the funding or sources of support (whether external or internal to your organization) received during this study, as detailed online in our guide for authors at http://journals.plos.org/plosone/s/submit-now.  Please also include the statement “There was no additional external funding received for this study.” in your updated Funding Statement.

ii) Please include your amended Funding Statement within your cover letter. We will change the online submission form on your behalf.

5. We note that you are reporting an analysis of a microarray, next-generation sequencing, or deep sequencing data set. PLOS requires that authors comply with field-specific standards for preparation, recording, and deposition of data in repositories appropriate to their field. Please upload these data to a stable, public repository (such as ArrayExpress, Gene Expression Omnibus (GEO), DNA Data Bank of Japan (DDBJ), NCBI GenBank, NCBI Sequence Read Archive, or EMBL Nucleotide Sequence Database (ENA)). In your revised cover letter, please provide the relevant accession numbers that may be used to access these data. For a full list of recommended repositories, see http://journals.plos.org/plosone/s/data-availability#loc-omics or http://journals.plos.org/plosone/s/data-availability#loc-sequencing.

Reviewers' comments:

Reviewer's Responses to Questions

**Comments to the Author**

1. Is the manuscript technically sound, and do the data support the conclusions?

Reviewer #1: Partly

Reviewer #2: Yes

2. Has the statistical analysis been performed appropriately and rigorously? 

Reviewer #1: Yes

Reviewer #2: Yes

3. Have the authors made all data underlying the findings in their manuscript fully available?

Reviewer #1: Yes

Reviewer #2: Yes

4. Is the manuscript presented in an intelligible fashion and written in standard English?

Reviewer #1: Yes

Reviewer #2: Yes

5. Review Comments to the Author

Reviewer #1: This manuscript describes studies on a novel lectin named TTRM_L1 that displays cellular toxicity toward a number of cell lines. The paper is preliminary in many ways, and should be significantly revised.

1. The specificity of cellular cytotoxicity should be expanded to include non-tumor cell lines, such as primary fibroblasts and human lymphocytes stimulated to divide.

2. The specificity of the lectin and its binding to glycans appears to be broad and certainly not to structures that are unique to tumor cells.

3. The authors should assess cell binding independently of cell killing using the conventional flow cytometric or fluorescent assays commonly used. Does the lectin bind to all cells tested?

4. Is binding and killing by the lectin inhibited by carbohydrate-based haptens? This is important, as it would provide more assurance that carbohydrate binding to cells is essential to cell killing observed.

5. The word hapten is not used, neither is carbohydrate inhibition, which is really unusual for a paper describing a lectin that binds to cells and kills them.

6. The mechanism of cell killing by the lectin is not clear, and the authors should provide specific evidence in that regard. Do the cells undergo apoptosis? The authors do not explicity state this one way or the other?

7. Does the lectin arrest cell division? Does the lectin cause cellular lysis? The authors also imply that the lectin inhibits the growth of a plant fungus, which is hardly a tumor cell. So the specificity of the lectin for killing tumor cells is a bit of a stretch for the title and the conclusions.

8. What are the mono/disaccharide haptens of the lectin? This is another common parameter typically assessed for novel lectins.

9. The name of the lectin used in this paper is awkward, in that it has an underline in it. Moreover, the name should be the name of the genus/species or some idea of the organism making it? The concept that it is merely cloned and expressed by an unknown organism further illustrates the preliminary nature of the paper.

Reviewer #2: The manuscript entitled "Discovery of antitumor lectins from competing microorganismal ecologic niches" by Lawanprasert et al. describes the discovery of a engineered lectin with selectivity for cancer carbohydrate antigens and cause cell death. Overall this is an excellent manuscript worthy of publication in PLOS ONE. The description of a novel very-potent anticancer protein is of significant interest. The paper could be improved with minor revision.

Specific comments below.

Introduction:

The introduction could be improved with better referencing of statements ( i.e "lectins are found in organismal proteomes, lipides and secretes" - referees for each category should be included).

Some of the language in the introduction is too casual. For example the opening sentence of paragraph 3 states " Nature contains a much broader diversity of lectins than present in any single species genome" - this is obvious and is followed by a statement on therapeutic potential that does not rely on the opening statement.

The abbreviation MBL should not be used in a general sense as it is generally used only to refer to mannose binding lectin in human plasma. It should not be used as an acronym for all mannose-specific lectins such as, I believe, griffithsin in one sentence followed by a reference to "the" MBL in the next sentence two references for different proteins are both being referred to by the same abbreviation.

the last sentence of paragraph 3 also seem a bit of a throw away line. No other reference to toxin conjugates and no real attachment to the rest of the paragraph.

Results:

The method by which the authors identified lectins in the genome should be discussed as this is not common knowledge.

The authors state that due to the varied species and interdependence of rainforest communities suggests that many of the 80 lectins identified "are" involved in between-species interactions. - There is no reason to expect so unless they can also show these proteins are secreted or are surface expressed. Stating that some of these proteins could be involved in such interactions seems more accurate.

The models produced for figure 2 are just that, speculative models. Paper could be significantly improved with actual structures.

Results could be improved with studies on agglutination and /or cell permeability changes associated with treating cells with TRRM1. Also confocal studies would be improved by including insensitive cells to see differences in TRRM location with time.

Figure 5 would be improved by including a legend fo monosaccharide identities for readers not familiar with the nomenclature of oligosaccharide structure.

paper could be improved by inclusion of direct binding studies ( with quantitation) between select oligosaccharides and TRRM1. ITC, SPR etc would be good.

Any evidence of aggregation in TRRM1? Is the expressed protein mono disperse? Some evidence of the purity and identity of the expressed protein would be useful to see at least in the supplemental data. LC/MS sequencing, SDS-Page...

Discussion:

The discussion should be better referenced. Several statements in the opening paragraph should be referenced.

The speculation in paragraph 2 on mechanism, based solely on the model of protein structure are weak. Better mechanism discussion would be achieved by additional experimental data on extrinsic vs. intrinsic pathways and data on apoptosis ve. necrosis, membrane binding differences between sensitive and resistant cells, etc.

Additional speculation on a ER-driven mechanism would be enhanced by better timelines of cytotoxicity and looking directly at ER flux in TRRM1 treated cells. Such experiments would improve this manuscript and the discussion.

Overall the manuscript is worthy of publication but could be improved by a few select experiments to better define mechanism and a a bit more data ( purity analysis of recombinant TRRM1, confocal controls with resistant cells).

6. PLOS authors have the option to publish the peer review history of their article (what does this mean?). If published, this will include your full peer review and any attached files.

Reviewer #1: No

Reviewer #2: No

---

## [Author Response · Author response to Decision Letter 0]

15 Jan 2020

See Response to Reviewers attachment.

---

## [Decision Letter · Decision Letter 1]

7 Feb 2020

Discovery of antitumor lectins from rainforest tree root transcriptomes

PONE-D-19-29568R1

Dear Dr. Medina,

We are pleased to inform you that your manuscript has been judged scientifically suitable for publication and will be formally accepted for publication once it complies with all outstanding technical requirements.

With kind regards,

Roger Chammas, M.D, Ph.D

Academic Editor

PLOS ONE

Additional Editor Comments (optional):

Reviewers' comments:

Reviewer's Responses to Questions

**Comments to the Author**

1. If the authors have adequately addressed your comments raised in a previous round of review and you feel that this manuscript is now acceptable for publication, you may indicate that here to bypass the “Comments to the Author” section, enter your conflict of interest statement in the “Confidential to Editor” section, and submit your "Accept" recommendation.

Reviewer #2: All comments have been addressed

2. Is the manuscript technically sound, and do the data support the conclusions?

Reviewer #2: (No Response)

3. Has the statistical analysis been performed appropriately and rigorously? 

Reviewer #2: Yes

4. Have the authors made all data underlying the findings in their manuscript fully available?

Reviewer #2: Yes

5. Is the manuscript presented in an intelligible fashion and written in standard English?

Reviewer #2: Yes

6. Review Comments to the Author

Reviewer #2: I believe that the authors have done an excellent job of addressing both my and the other reviewer's comments and suggestions. I believe that the manuscript has been significantly improved and is now suitable for publication in PLOS ONE.

7. PLOS authors have the option to publish the peer review history of their article (what does this mean?). If published, this will include your full peer review and any attached files.

Reviewer #2: No

---

## [Editor Report · Acceptance letter]

12 Feb 2020

PONE-D-19-29568R1 

Discovery of antitumor lectins from rainforest tree root transcriptomes 

Dear Dr. Medina:

I am pleased to inform you that your manuscript has been deemed suitable for publication in PLOS ONE. Congratulations! Your manuscript is now with our production department. 

With kind regards,

on behalf of

Prof. Roger Chammas 

Academic Editor

PLOS ONE